# Unimproved source of drinking water and the associated factors: Insights from the 2020 Somalia demographic and health survey

**Abdisalam Mahdi Hassan**[1], **Nimo Mohamoud Barakale**[1], **Omran Salih**[2]*, **Abdisalam Hassan Muse**[1]

**1** School of Postgraduate Studies and Research, Amoud University, Borama, Somaliland, **2** Institute of Systems Science, Durban University of Technology, Durban, South Africa

* omran.salih@amoud.edu.so

**Data Availability Statement:** This study used secondary data from the Somalia Health and Demographic Survey (SHDS) 2020, which is publicly available on the website (https://microdata.

## Abstract

Access to safe drinking water is a fundamental human right and a critical public health concern, particularly in lower- and middle-income countries with limited infrastructure. Somalia faces significant challenges in providing improved drinking water sources, with a high prevalence of unimproved sources. This study analyzes data from the SHDS 2020 to investigate the prevalence of unimproved drinking water sources and identify associated factors. A cross-sectional study of 32,300 participants was conducted to identify factors associated with using unimproved drinking water sources. Multivariable logistic regression analysis was performed using Stata 16 software. Variables with a p-value < 0.05 in bivariate analysis were included in the multivariate model. Adjusted odds ratios (AOR) and 95% confidence intervals (CI) were used to estimate the association of significant variables with the outcome. Approximately 22.04% (95% CI: 21.5%, 22.4%) of the Somali population utilizes unimproved drinking water sources. Households with a head aged 20 years and above (AOR = 0.88, p = 0.059) were less likely to rely on unimproved sources than households with a head under 20 years. Female household heads (AOR = 1.17, p = 0.000) were more likely to use unimproved sources than male household heads. Unemployed partners (AOR = 1.14, p = 0.000) were more likely to use unimproved sources than employed partners. Rural (AOR = 1.12, p = 0.013) and nomadic (AOR = 0.93, p = 0.175) residents were more likely to use unimproved sources than urban residents. Households in Mudug (AOR = 31.18, p = 0.000), Nugaal (AOR = 4.15, p = 0.000), Bari (AOR = 5.26, p = 0.000), and Sanaag (AOR = 2.52, p = 0.000) regions were less likely to use unimproved sources compared to households in other regions. These findings highlight the urgent need for Somalia to improve its provision of safe and accessible water sources.

## Introduction

Access to clean water is a fundamental human right, essential for human development, growth, and well-being [1]. It is a crucial element in achieving sustainable development, as highlighted

nbs.gov.so/index.php/catalog/50). Access was granted after registration using an institutional email (abdisalam.hassan@amoud.edu.so). As the data is publicly available and anonymized, formal ethical approval from an Institutional Review Board (IRB) or Ethics Committee was not required for our study. However, it is important to acknowledge the ethical considerations during the original SHDS 2020 data collection. The National Bureau of Statistics (NBS), responsible for conducting the SHDS, likely has an established IRB or Ethics Committee that reviewed and approved the survey design and data collection procedures. The statement mentions that enumerators received training on obtaining informed consent (likely verbal) from participants, ensuring confidentiality, and building rapport. This aligns with ethical research practices for human subjects. Further details on the original SHDS 2020 IRB approval process or specific consent procedures could be obtained by contacting the NBS.

**Funding:** The authors received no specific funding for this work

**Competing interests:** The authors have declared that no competing interests exist.

by Sustainable Development Goal (SDG) 6, which explicitly addresses water and sanitation with the target of providing universal and equitable access to safe and affordable drinking water for all by 2030 [2]. The long-term objective is to ensure that everyone has access to safe water at home, ideally within a 30-minute round trip [3, 4]. However, globally, over 2 billion people still lack access to safely managed drinking water services [5].

The burden of inadequate water access disproportionately affects low- and middle-income countries, particularly in sub-Saharan Africa, where infrastructure limitations, poverty, and conflict often impede the provision of safe water sources [3]. The majority of households with unimproved water sources are found in sub-Saharan Africa, often requiring long journeys to collect water [3]. This lack of safe water access contributes to increased risks of infectious diseases, including cholera, typhoid, schistosomiasis, and infections of the respiratory, skin, and eye systems [1].

Somalia, a country located in sub-Saharan Africa, faces significant challenges in providing safe drinking water to its population [6–8]. Despite the national target of achieving universal access to an improved source of drinking water outlined in the Somalia National Development Plan (NDP), the population in Somalia has not yet been able to reach this goal [5, 9, 10]. In Somalia, slightly over three-quarters (76%) of urban households have access to improved water sources, compared to just over half (55%) of rural households and 35% of nomadic households [11]. This disparity in water access has serious implications for the health and well-being of Somali communities, leading to increased vulnerability to waterborne diseases and hindering progress towards achieving national development goals.

The lack of comprehensive research on the prevalence and associated factors of unimproved drinking water sources in Somalia represents a significant research gap [3]. Existing data suggests that 35% of Somali households rely on unimproved sources for drinking water [11], highlighting the need for a deeper understanding of this issue. Previous studies have examined factors influencing water access in other African countries, providing valuable insights into the complex factors at [12–16]. However, a comprehensive investigation using nationally representative data on unimproved sources in Somali households is lacking.

This study aims to fill this critical research gap by analyzing data from the 2020 Somalia Health and Demographic Survey (SHDS) to investigate the prevalence of unimproved drinking water sources and identify associated factors. The study's findings could assist policymakers in implementing measures to reduce the consequences of utilizing water from unimproved sources in Somalia and work towards achieving the national goal of universal access to safe water.

## Material and methods

### Study design

This cross-sectional study utilized secondary data from the 2020 Somalia Health and Demographic Survey (SHDS), the first nationally representative household survey conducted in Somalia. The SHDS is a crucial resource for strengthening national data systems and promoting evidence-based planning for improved public health outcomes.

### Study setting

The study focused on Somalia, a country located in the Horn of Africa with an estimated population of 12.3 million. Somalia boasts the longest coastline in Africa, stretching 3,333 kilometers along the Indian Ocean and the Gulf of Aden. It shares borders with Ethiopia to the west, Kenya to the southwest, and Djibouti to the northwest.

## Data source

This study utilized the 2020 SHDS dataset, collected between 2018 and 2020. The data was obtained from the DHS website: www.dhsprogram.com. Variables relevant to the study were extracted from the household record (HR file) dataset. The SHDS sample was specifically designed to represent key indicators across various strata within Somalia, achieving a response rate of 99%. This high response rate, encompassing urban, rural, and nomadic areas, reflects the willingness and cooperation of participants. The final sample size for this study, after data cleaning and the removal of missing variables, was 32,300 participants [7, 8, 17].

## Study variables

**Dependent variable.**   The outcome variable for this study is the utilization of unimproved sources of drinking water. The drinking water source was classified as unimproved if a household gets drinking water from an unprotected dug well, unprotected spring, surface water, and tanker truck/cart with a small tank. For more clarification, to define improved and unimproved drinking water sources, we utilize the classification system employed in the DHS dataset. A household's primary drinking water source is categorized as improved if it falls within the following categories: piped into dwelling or yard, public tap, standpipe, piped to neighbor, tube well/borehole, protected well, protected spring, rainwater collection, tanker truck/cart, or bottled water. These sources generally indicate a greater level of infrastructure, protection from contamination, and hygiene. Conversely, unimproved sources, categorized as unprotected well, unprotected spring, or "other," are more susceptible to contamination due to the lack of protective measures. It is important to note that even within these categories, the actual quality of the water can vary depending on the specific context, level of maintenance, and overall hygiene practices. For the purposes of our analysis, we utilize this classification system from the DHS dataset, acknowledging its inherent limitations and the potential for variation in water quality within these broad categories.

**Independent variables.**   Individual and community-level independent variables were considered in this study. The age of the household head (less than 20 years or 20 years and more), husband's education level, whether he attended school or not (yes or no) level of education of the respondent (no education, primary, secondary and higher), gender of the household head (male or female), household head employment status (yes or no) and the employment status of the husband/partner during the previous year was included as an individual-level factor in this study. Community-level factors, residence (urban, rural or nomadic), region (Awdal, Waqooyi galbeed, Togdheer, Sool, Sanaag, Bari, Nugaal, Mudug, Galgaduud, Hiraan, Middle shabelle, Banadir, Bay, Bakool, Gedo and Lower juba) wealth index of the family (lowest, second, middle, fourth and highest), family size members (less than four members and four or more members) and type of sanitation facility (improved or unimproved) were considered.

## Data management and statistical analysis

Data analysis was conducted using Stata version 16. Following data cleaning and coding, descriptive analysis was performed to determine the frequency and percentages of variables of interest. Due to the binary nature of the outcome variable (improved vs. unimproved water sources), bivariate analysis utilizing chi-square testing was employed to explore associations between the outcome variable and independent variables. Subsequently, multivariate binary logistic regression was conducted to assess the relationship between the outcome variable and a set of independent variables. Statistical significance was determined using an adjusted odds ratio (AOR) with a 95% confidence interval (CI) and a p-value threshold of 0.05.

### Ethical consideration

This study utilizes publicly available data from the Somalia Demographic and Health Survey (DHS) program. The data, obtained from the official Somalia DHS website, has been anonymized and contains no personal identifiers. As such, ethical approval was not required for this research. We adhere to the ethical principles of respecting data privacy and ensuring the anonymity of individuals.

## Results

### Exploratory analysis

The study's findings, based on data from the 2020 Somalia Health and Demographic Survey (SHDS), revealed several key factors associated with the use of unimproved drinking water sources. Univariate analysis (*Table 1*) showed that a significant proportion of household heads (80.64%) were aged 20 years and older. Additionally, a large percentage of respondents (87.02%) lacked formal education. The majority of household heads were male (66.40%), and a small percentage were employed (1.14%). Respondents resided in urban (41.35%), rural (27.29%), and nomadic (31.36%) areas. Notably, wealth status varied significantly among households, with the lowest (24.04%) and highest (16.46%) wealth indices representing the extremes. Most households (77.30%) consisted of four or more members, and a substantial proportion (59.70%) utilized unimproved sanitation facilities.

### Correlational analysis

Table 2 presents the bivariate analysis of factors associated with drinking water sources using the SHDS 2020 data. The results reveal several statistically significant relationships between explanatory variables and the use of unimproved water sources. Notably, younger respondents (under 20 years) were significantly more likely to access unimproved water sources ($\chi^2$ = 996.47, p < .001). Higher levels of education were significantly associated with a lower likelihood of using unimproved water sources ($\chi^2$ = 17.67, p < .001). Male household heads were significantly more likely to use unimproved water sources compared to female heads ($\chi^2$ = 45.76, p < .001). Similarly, unemployed household heads ($\chi^2$ = 22.09, p < .001), partners without employment ($\chi^2$ = 203.58, p < .001), residents of rural areas ($\chi^2$ = 516.12, p < .001) and nomadic areas ($\chi^2$ = 4200, p < .001) were all significantly more likely to access unimproved water sources. Households with the highest wealth status were significantly more likely to use unimproved water sources compared to those in the middle and lower wealth strata ($\chi^2$ = 164.04, p < .001). Finally, larger families were less likely to use unimproved water sources ($\chi^2$ = 5.51, p < .02).

### Regression analysis

*Table 3* presents the results of multivariable binary logistic regression analysis, examining the association between individual and community-level factors and unimproved drinking water sources. Household heads aged 20 years and older were less likely to use unimproved water sources compared to those under 20 years (AOR = 0.88, 95% CI = 0.77, 1.00, p = .059). Respondents with secondary or higher education levels were more likely to use unimproved water sources compared to those with no education or primary education (AOR = 1.32, 95% CI = 0.90, 1.93, p = .128). Female household heads were more likely to use unimproved water sources compared to male heads (AOR = 1.18, 95% CI = 1.10, 1.25, p < .001). Unemployed household heads were also more likely to access unimproved water sources compared to employed household heads (AOR = 1.14, 95% CI = 0.89, 1.44, p = .276). Partners without

**Table 1. Univariate analysis of both individual and community level factors associated with source of drinking water using SHDS 2020.** (n = 32300).

| Variable | Categories | Frequency (n) | Percentage % |
|---|---|---|---|
| Age of household head | Less than 20 years | 6253 | 19.36 |
|  | 20 years and more | 26047 | 80.64 |
| Head of household education level | Yes | 26985 | 83.54 |
|  | No | 5315 | 16.46 |
| Level of education of the respondent | No education | 28108 | 87.02 |
|  | Primary education | 3229 | 10 |
|  | Secondary education | 796 | 2.46 |
|  | Higher education | 167 | 0.52 |
| Gender of household head | Male | 21447 | 66.40 |
|  | Female | 10853 | 33.60 |
| Household head employment status | Yes | 367 | 1.14 |
|  | No | 31933 | 98.86 |
| Partner work | Yes | 13985 | 43.30 |
|  | No | 18315 | 56.70 |
| Residence | Urban | 13357 | 41.35 |
|  | Rural | 8814 | 27.29 |
|  | Nomadic | 10129 | 31.36 |
| Region | Awdal | 1333 | 4.13 |
|  | Waqoyigalbed | 2299 | 7.12 |
|  | Togdheer | 2798 | 8.66 |
|  | Sool | 2733 | 8.46 |
|  | Sanaag | 2727 | 8.44 |
|  | Bari | 1274 | 3.94 |
|  | Nugaal | 1498 | 4.64 |
|  | Mudug | 1490 | 4.61 |
|  | Galgadud | 1852 | 5.73 |
|  | Hiran | 2131 | 6.60 |
|  | Middle shabele | 1564 | 4.84 |
|  | Banadir | 4154 | 12.86 |
|  | Bay | 789 | 2.44 |
|  | Bakool | 1670 | 5.17 |
|  | Gedo | 1895 | 5.87 |
|  | Lower juba | 2093 | 6.48 |
| Wealth index of the family | Lowest | 7766 | 24.04 |
|  | Second | 7352 | 22.76 |
|  | Middle | 6045 | 18.72 |
|  | Fourth | 5819 | 18.02 |
|  | Highest | 5318 | 16.46 |
| Family size | Less than four members | 7331 | 22.70 |
|  | Four members or more | 24969 | 77.30 |
| Type of sanitation facility | Improved | 13016 | 40.30 |
|  | Unimproved | 19284 | 59.70 |

employment were more likely to use unimproved water sources compared to employed partners (AOR = 1.14, 95% CI = 1.07, 1.21, p < .001). Households in rural and nomadic areas were more likely to use unimproved water sources compared to those in urban areas (AOR = 1.12, 95% CI = 1.02, 1.23, p = .013; AOR = 0.93, 95% CI = 0.83, 1.03, p = .175). Households in the

**Table 2. Bivariate analysis for factors associated with source of drinking water in Somalia using SDHS 2020.**

| Variable | Categories | Source of drinking water | | Chi-square | Df | P-value |
|---|---|---|---|---|---|---|
| | | Improved | Unimproved | | | |
| Age of household head | Less than 20 years | 5804 (92.82) | 449 (7.18) | 996.4742 | 1 | 0.000 |
| | 20 years and more | 19377 (74,39) | 6670 (25.61) | | | |
| Head of household education level | Yes | 21132 (78.31) | 5853 (21.69) | 11.7196 | 1 | 0.001 |
| | No | 4049 (76.18) | 1266 (23.82) | | | |
| Level of education of the respondent | No education | 21928 (78.01) | 6180 (21.99) | 17.6716 | 3 | 0.001 |
| | Primary education | 2552 (79.03) | 677 (20.97) | | | |
| | Secondary education | 576 (72.36) | 220 (27.64) | | | |
| | Higher education | 125 (74.85) | 42 (25.15) | | | |
| Gender of household head | Male | 16482 (76.85) | 4965 (23.15) | 45.7568 | 1 | 0.000 |
| | Female | 8699 (80.15) | 2154 (19.85) | | | |
| HH head employment status | Yes | 249 (67.85) | 118 (32.15) | 22.0925 | 1 | 0.000 |
| | No | 24932 (78.08) | 7001 (21.92) | | | |
| Partner work | Yes | 10376 (74.19) | 3609 (25.81) | 203.5755 | 1 | 0.000 |
| | No | 14805 (80.84) | 3510 (19.16) | | | |
| Residence | Urban | 9581 (71.73) | 3776 (28.27) | 516.1233 | 2 | 0.000 |
| | Rural | 7296 (82.78) | 1518 (17.22) | | | |
| | Nomadic | 8304 (81.98) | 1825 (18.02) | | | |
| Region | Awdal | 1077 (80.80) | 256 (19.20) | 4200 | 15 | 0.000 |
| | Waqoyi-galbed | 2026 (88.13) | 273 (11.87) | | | |
| | Togdheer | 2167 (77.45) | 631 (22.55) | | | |
| | Sool | 2415 (88.36) | 318 (11.64) | | | |
| | Sanaag | 2542 (93.22) | 185 (6.78) | | | |
| | Bari | 1225 (96.15) | 49 (3.85) | | | |
| | Nugaal | 1440 (96.13) | 58 (3.87) | | | |
| | Mudug | 1483 (99.53) | 7 (0.47) | | | |
| | Galgaduud | 1601 (86.45) | 251 (13.55) | | | |
| | Hiraan | 1839 (86.30) | 292 (13.70) | | | |
| | Middle-shabele | 919 (58.76) | 645 (41.24) | | | |
| | Banadir | 2274 (54.74) | 1880 (45.26) | | | |
| | Bay | 533 (67.55) | 256 (32.45) | | | |
| | Bakool | 1127 (67.49) | 543 (32.51) | | | |
| | Gedo | 1307 (68.97) | 588 (31.03) | | | |
| | Lower-juba | 1206 (57.62) | 887 (42.38) | | | |
| Wealth index of the family | Lowest | 6451 (83.07) | 1315 (16.93) | 164.0400 | 4 | 0.000 |
| | Second | 5654 (76.90) | 1698 (23.10) | | | |
| | Middle | 4658 (77.06) | 1387 (22.94) | | | |
| | Fourth | 4429 (76.11) | 1390 (23.89) | | | |
| | Highest | 3989 (75.01) | 1329 (24.99) | | | |
| Family size | Less than four members | 5642 (76.96) | 1689 (23.04) | 5.5071 | 1 | 0.019 |
| | Four members or more | 19539 (78.25) | 5430 (21.75) | | | |
| Type of sanitation facility | Unimproved | 13194 (68.42) | 6090 (31.58) | 2500 | 1 | 0.000 |
| | Improved | 11987 (92.09) | 1029 (7.91) | | | |

Mudug, Nugaal, Bari, and Sanaag regions were also more likely to use unimproved water sources compared to households in other regions (AOR = 31.18, 95% CI = 14.62, 66.52, p < .001; AOR = 4.15, 95% CI = 3.07, 5.61, p < .001; AOR = 5.26, 95% CI = 3.82, 7.24, p < .001;

**Table 3. Multivariable logistic regression analysis for individual and community level factors associated with unimproved source of drinking water using SHDS 2020 data.**

| Variables | Categories | AOR | Coefficient (S.E) | 95% CI | P-value |
|---|---|---|---|---|---|
| Age of household head | Less than 20 years | Ref | | | |
| | 20 years and more | .883176 | .0579984 | .7765129 1.004491 | 0.059 |
| Head of household education level | Yes | Ref | | | |
| | No | 1.097976 | .0480014 | 1.007813 1.196205 | 0.033 |
| Level of education of the respondent | No education | Ref | | | |
| | Primary education | .9066157 | .0474836 | .818167 1.004626 | 0.061 |
| | Secondary education | 1.04202 | .0966024 | .8688877 1.249649 | 0.657 |
| | Higher education | 1.322392 | .2563345 | .9044029 1.933563 | 0.657 |
| Gender of household head | Male | Ref | | | |
| | Female | 1.176338 | .0374891 | 1.105108 1.252159 | 0.000 |
| HH head employment status | Yes | Ref | | | |
| | No | 1.14179 | .13898 | .8994495 1.449424 | 0.276 |
| Partner work | Yes | Ref | | | |
| | No | 1.143356 | .0369817 | 1.073123 1.218185 | 0.000 |
| Residence | Urban | Ref | | | |
| | Rural | 1.123391 | .0527323 | 1.02465 1.231648 | 0.013 |
| | Nomadic | .9277132 | .0513651 | .8323096 1.034052 | 0.175 |
| Region | Awdal | Ref | | | |
| | Waqoyi-galbed | 1.725367 | .1663881 | 1.428219 2.084338 | 0.000 |
| | Togdheer | .7651895 | .0649384 | .6479346 .9036639 | 0.002 |
| | Sool | 1.332351 | .1266288 | 1.105908 1.60516 | 0.003 |
| | Sanaag | 2.521921 | .2667799 | 2.049688 3.102954 | 0.000 |
| | Bari | 5.255723 | .8575258 | 3.817245 7.236271 | 0.000 |
| | Nugaal | 4.145597 | .6379567 | 3.066179 5.605012 | 0.000 |
| | Mudug | 31.18253 | 12.05507 | 14.61637 66.52473 | 0.000 |
| | Galgaduud | 1.146988 | .1154424 | .9416438 1.39711 | 0.173 |
| | Hiraan | 1.348201 | .130061 | 1.115936 1.628809 | 0.002 |
| | Middle-shabele | .3683359 | .0332356 | .3086303 .4395918 | 0.000 |
| | Banaadir | .3903689 | .032654 | .3313394 .4599149 | 0.000 |
| | Bay | .6709773 | .0727668 | .5424951 .8298886 | 0.000 |
| | Bakool | .6522254 | .0587195 | .5467194 .7780918 | 0.000 |
| | Gedo | .648793 | .0570891 | .5460175 .7709138 | 0.000 |
| | Lower-juba | .4060782 | .0349571 | .3430318 .4807121 | 0.000 |
| Wealth index of the family | Lowest | Ref | | | |
| | Second | 1.016886 | .0483318 | .9264354 1.116167 | 0.725 |
| | Middle | 1.121843 | .0668487 | .9981838 1.260822 | 0.054 |
| | Fourth | 1.027572 | .0649364 | .9078649 1.163062 | 0.667 |
| | Highest | .9914159 | .0655665 | .870888 1.128624 | 0.896 |
| Family size | Less than four members | Ref | | | |
| | Four members or more | .9823335 | .0339963 | .9179116 1.051277 | 0.607 |
| Type of sanitation facility | Improved | 2.220988 | .110268 | 2.015049 2.447974 | 0.000 |
| | Unimproved | 2.706836 | .4468044 | 1.958656 3.74081 | 0.000 |

AOR = 2.52, 95% CI = 2.05, 3.10, p < .001). There was no significant association between wealth index and the use of unimproved water sources. Finally, households with four or more members were less likely to use unimproved water sources compared to households with fewer members (AOR = 0.98, 95% CI = 0.92, 1.05, p = .607).

## Discussion

This study, utilizing data from the 2020 Somalia Health and Demographic Survey, provides valuable insights into the prevalence and associated factors of unimproved drinking water sources in Somalia. Our findings highlight a significant prevalence of 22.04% of households relying on unimproved water sources, exceeding proportions reported in some other African countries, such as Ghana, while falling below those observed in Ethiopia and Eswatini. These disparities likely stem from variations in economic status, literacy rates, study periods, and specific study settings across these nations [1, 12, 18, 19].

The study partially aligns with prior research, confirming the significant role of individual and community-level factors in determining access to improved water sources. Consistent with studies in Ethiopia and Ghana, we found that higher levels of education are associated with a lower likelihood of using unimproved water sources, likely due to increased awareness of hygiene and sanitation practices [20, 21]. Similarly, we observed a correlation between wealth index and water source utilization, with lower income households being more likely to rely on unimproved sources, as observed in studies conducted in Ethiopia [3]. This finding underscores the importance of economic empowerment in promoting access to improved water sources.

However, the findings contradict some previously observed patterns. We found a significant association between female-headed households and the use of unimproved water sources, contrary to findings in Ghana [12]. This discrepancy may indicate specific challenges faced by female-headed households in Somalia, potentially due to limited economic resources or societal constraints, warranting further exploration. Additionally, our results demonstrate a significant association between unemployed partners and the use of unimproved water sources, suggesting that lack of employment may hinder access to improved water sources due to financial constraints.

A key contribution of this study lies in its comprehensive examination of factors associated with unimproved drinking water sources within the context of Somalia, using nationally representative data. This provides crucial information for policymakers and stakeholders to develop targeted interventions and policies to improve water access in the country. By identifying specific vulnerable populations, such as households headed by females, those with unemployed partners, and those residing in rural and nomadic areas, the study provides a framework for tailored strategies.

This study is particularly relevant in the context of Sustainable Development Goal (SDG) 6, which emphasizes the need to ensure access to safe and affordable drinking water for all by 2030 [2]. By highlighting the significant prevalence of unimproved water sources in Somalia and identifying associated factors, the study underscores the urgent need for targeted interventions to achieve SDG 6 in Somalia. This includes initiatives to improve water infrastructure, promote sanitation practices, and address socio-economic factors hindering access to clean water, particularly for vulnerable populations.

Despite its strengths, this study has limitations. First, due to the cross-sectional design, it does not establish causality between the observed factors and the use of unimproved water sources. Longitudinal studies are needed to further investigate the directionality of these relationships. Second, the study relies on self-reported data, which may be subject to recall bias. Further investigation using objective measures of water source quality and access would enhance the study's findings. Finally, the study focuses solely on access to water, neglecting other important factors such as water quality and sanitation, which are crucial for maintaining public health [1].

Overall, this study provides essential insights into the challenges and opportunities associated with access to clean water in Somalia. The findings highlight the need for multi-sectoral

interventions to improve water infrastructure, promote hygiene and sanitation practices, and address socio-economic inequalities, particularly for vulnerable populations. By focusing on these factors, Somalia can work towards achieving the national goal of universal access to safe water and contribute to the achievement of SDG 6.

## Conclusion

This study, using data from the 2020 Somalia Health and Demographic Survey (SHDS), revealed that approximately 22% of the Somali population utilizes unimproved drinking water sources. Our findings demonstrate that individual factors such as age of the household head, gender of the household head, education level, partner's attendance of school, and occupation status, along with community-level factors like residence, region, wealth index, family size, and type of sanitation facility, significantly influence the use of unimproved water sources. These results highlight the multifaceted nature of this issue, underscoring the need for multi-sectoral approaches to address the challenge of inadequate access to safe drinking water in Somalia.

## Policy implications

The findings of this study provide valuable insights for policymakers and stakeholders in Somalia to develop targeted interventions and policies aimed at improving access to safe drinking water and promoting sustainable development.

1. Prioritize Vulnerable Groups: Policymakers should prioritize interventions for vulnerable populations, such as female-headed households, individuals with unemployed partners, and residents of rural and nomadic areas. This includes targeted programs to improve water infrastructure in these communities, provide financial assistance for water access improvements, and promote awareness of hygiene and sanitation practices.

2. Promote Education and Economic Empowerment: Investing in education and economic empowerment initiatives can contribute to a long-term solution to the problem of unimproved water sources. Educated individuals are more likely to understand the importance of clean water and sanitation practices, and economic empowerment allows households to afford improved water sources.

3. Strengthen Water Infrastructure: Investing in and maintaining water infrastructure is crucial for expanding access to safe drinking water. This includes constructing new water supply systems, rehabilitating existing ones, and promoting efficient water management practices.

4. Foster Multi-Sectoral Collaboration: Addressing the challenges of unimproved water sources requires collaboration across various sectors, including government, civil society, and private sector. This collaboration is essential for developing comprehensive and effective solutions.

5. Monitor and Evaluate Progress: Regular monitoring and evaluation of interventions is crucial to track progress and ensure that programs are effective in reaching their goals.

By focusing on these policy implications, Somalia can work towards achieving the national goal of universal access to safe water, contributing to the achievement of SDG 6 and improving the overall well-being of its citizens.

## Author Contributions

**Conceptualization:** Abdisalam Mahdi Hassan, Abdisalam Hassan Muse.

**Formal analysis:** Nimo Mohamoud Barakale.

**Methodology:** Nimo Mohamoud Barakale, Abdisalam Hassan Muse.

**Supervision:** Abdisalam Mahdi Hassan, Omran Salih, Abdisalam Hassan Muse.

**Validation:** Omran Salih, Abdisalam Hassan Muse.

**Visualization:** Omran Salih.

**Writing – original draft:** Nimo Mohamoud Barakale.

**Writing – review & editing:** Omran Salih, Abdisalam Hassan Muse.

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
