## [Decision Letter · Decision Letter 0]

3 Sep 2024

PGPH-D-24-01613

Unimproved Source of Drinking Water and its Associated Factors: Insights from Somalia Health and Demographic Survey 2020 Data

Dear Dr. Salih,

Thank you for submitting your manuscript to PLOS Global Public Health. After careful consideration, we feel that it has merit but does not fully meet PLOS Global Public Health’s publication criteria as it currently stands. Therefore, we invite you to submit a revised version of the manuscript that addresses the points raised during the review process.

Please you are implored to critically consider each and every comment provided by the reviewers and revise your manuscript accordingly. Very importantly, make sure to highlight any changes to the revised draft for further assessment.

We look forward to receiving your revised manuscript.

Kind regards,

Razak M Gyasi, PhD, PD

Academic Editor

Journal Requirements:

Additional Editor Comments (if provided):

Reviewers' comments:

Reviewer's Responses to Questions

**Comments to the Author**

1. Does this manuscript meet PLOS Global Public Health’s publication criteria? Is the manuscript technically sound, and do the data support the conclusions? The manuscript must describe methodologically and ethically rigorous research with conclusions that are appropriately drawn based on the data presented.

Reviewer #1: Yes

Reviewer #2: No

Reviewer #3: Partly

Reviewer #4: Yes

Reviewer #5: Yes

Reviewer #6: Yes

2. Has the statistical analysis been performed appropriately and rigorously?

Reviewer #1: Yes

Reviewer #2: N/A

Reviewer #3: Yes

Reviewer #4: Yes

Reviewer #5: Yes

Reviewer #6: Yes

3. Have the authors made all data underlying the findings in their manuscript fully available (please refer to the Data Availability Statement at the start of the manuscript PDF file)?

Reviewer #1: Yes

Reviewer #2: No

Reviewer #3: Yes

Reviewer #4: Yes

Reviewer #5: Yes

Reviewer #6: Yes

4. Is the manuscript presented in an intelligible fashion and written in standard English?

Reviewer #1: Yes

Reviewer #2: No

Reviewer #3: Yes

Reviewer #4: Yes

Reviewer #5: Yes

Reviewer #6: Yes

5. Review Comments to the Author

Reviewer #1: The manuscript has no issues about dual publication, research ethics, or publication ethics however there are some minor issues to be addressed. The author should address the following issues:

1. The definition or criteria for improved and unimproved sanitation facilities

2. The author indicated that variables with p-values less than 0.4 in the multivariate model were fitted in the bivariate analysis. Why was less than 0.4 was used?

3. The author should address a few grammatical error issues. For instance, was is used instead of were, Can have got more, etc.)

4.Six (6) references are not enough for such research.

Reviewer #2: The study has a commendable objective of attempting to explore factors associated with unimproved drinking water sources in Somalia.

However, some of the major concerns with the submitted manuscript include:

1. It is not well formatted. The manuscript has sections that needs to be rewritten for clarity.

2. The discussion section is not thoroughly informed by the presented exploratory analysis.

3. The study does not sufficiently engage existing literature or previous works (or may have not accurately cited all of the sources used) hence only 6 articles are referenced in the reference section.

4. The study does not sufficiently address practical implications or recommendations for its discussed findings.

Reviewer #3: Original and pertinent work using robust methodology.

The recommendation in concluding section is generic and does not clearly link to study findings.

Both authors state contribution to data the collection, but it’s not clear if they participated in primary study.

Authors should verify, ethical approval process of the primary study and affirm that standard ethical principles were followed rather than an assumption that it “likely” followed the same.

Reviewer #4: Manuscript is well coordinated however there are a few points to consider.Please refer to review submitted. Please re look at manuscript for grammatical errors and areas where lowercase and uppercase letters has been used.

Reviewer #5: The manuscript presents an important topic in the field of global public health. This document outlines a study examining the factors associated with the use of unimproved drinking water sources in Somalia, based on data from the 2020 Somalia Health and Demographic Survey (SHDS). The study highlights that approximately 22.04% of the Somali population relies on unimproved water sources, with significant disparities based on household characteristics, such as age, education level, residence, region, wealth index, and employment status.

The manuscript is generally well-written, but there are areas where clarity could be improved. Some sections, particularly the methods and discussion, are dense and could benefit from more concise language. Additionally, some figures and tables are difficult to interpret and would be more effective with clearer labels and explanations.

The manuscript briefly mentions ethical approval, but there is no discussion of how informed consent was obtained from participants. Given the sensitive nature of the study topic, it is crucial to provide more information on the ethical procedures followed to ensure participant safety and confidentiality.

I recommend the authors address the methodological issues raised, particularly regarding the sample size and data analysis. A more cautious interpretation of the results would also strengthen the manuscript. Clarifying the ethical considerations and improving the presentation of data will help ensure the study's impact on the field.

Reviewer #6: Feedback:

1. Clarity and Structure:

o Introduction: The introduction is somewhat cluttered with information and could benefit from clearer organization. The background information on global water access and the SDG goals is useful, but the connection to Somalia’s specific context could be strengthened. The narrative would be clearer if it flowed from the global context to the specific challenges in Somalia, leading into the study’s objectives.

o Results Section: The presentation of results is dense and could be streamlined for better readability. The inclusion of too many details in the text, such as the exact percentages and odds ratios is overwhelming. Consider summarizing key findings in a more digestible format, perhaps using bullet points or summary tables for major results, and focusing the narrative on interpreting these findings.

2. Interpretation of Findings:

o The discussion does not fully explore the implications of the findings. For example, the higher likelihood of unimproved water sources in female-headed households is noted, but the underlying reasons for this disparity are not sufficiently explored. Additionally, the manuscript could benefit from a more in-depth discussion on why certain regions, such as Mudug and Nugaal, show different patterns compared to others.

o The conclusion mentions that Somalia should enhance its provision of improved water sources but does not provide specific recommendations or strategies on how this could be achieved, given the identified factors.

3. Inconsistencies in Reporting:

o There are some inconsistencies in the reporting of results, particularly in the regression analysis. For instance, the manuscript states that "the odds of unemployed household heads are 1.14 times more likely to use unimproved sources of drinking water," but it is unclear whether this association is statistically significant. Clearer reporting of significance levels and confidence intervals throughout the results section would improve clarity.

4. Lack of Policy Implications:

o While the study identifies several factors associated with the use of unimproved water sources, the practical implications of these findings for policy and intervention strategies are not well developed. The paper would be stronger if it included more detailed recommendations for policymakers on how to address these issues, particularly in the Somali context.

Recommendations:

1. Reorganize the Introduction: Streamline the introduction to provide a clearer narrative that connects global water access issues to the specific challenges in Somalia, leading naturally into the study's objectives.

2. Enhance the Discussion: Expand on the discussion of findings, particularly regarding the socio-cultural and economic factors that may explain the disparities in water access. Provide more detailed policy recommendations based on the study’s findings.

3. Simplify the Results Presentation: Consider summarizing the key findings in a more accessible format and focusing the narrative on the interpretation rather than the granular details of the data.

4. Improve Reporting Consistency: Ensure that all significant results are clearly reported with appropriate p-values and confidence intervals. Address any inconsistencies in the statistical reporting.

5. Expand Policy Implications: Develop a more robust section on the practical implications of the findings, offering specific recommendations for how Somalia can improve water access, taking into account the identified factors.

6. PLOS authors have the option to publish the peer review history of their article (what does this mean?). If published, this will include your full peer review and any attached files.

**Do you want your identity to be public for this peer review?** For information about this choice, including consent withdrawal, please see our Privacy Policy.

Reviewer #1: **Yes: **FELIX GUMAAYIRI AABEBE

Reviewer #2: No

Reviewer #3: **Yes: **Bonaventure Ahaisibwe MBChB , MPH

Reviewer #4: No

Reviewer #5: **Yes: **AYA AKSH

Reviewer #6: No

---

## [Decision Letter · Decision Letter 1]

5 Nov 2024

PGPH-D-24-01613R1

Unimproved Source of Drinking Water and its Associated Factors: Insights from Somalia Health and Demographic Survey 2020.

Dear Dr. Salih,

Thank you for submitting your manuscript to PLOS Global Public Health. After careful consideration, we feel that it has merit but does not fully meet PLOS Global Public Health’s publication criteria as it currently stands. Therefore, we invite you to submit a revised version of the manuscript that addresses the points raised during the review process.

We look forward to receiving your revised manuscript.

Kind regards,

Professor Razak Gyasi, PhD, PD

Academic Editor

Journal Requirements:

Additional Editor Comments (if provided):

Please, review the attachment by the Reviewer thoroughly and revise the manuscript accordingly. Very importantly, please revise the manuscript for language correctness and critically improve the English structure.

Revise the title as follows: Unimproved Source of Drinking Water and the Associated Factors: Insights from the 2020 Somalia Demographic and Health Survey, instead of the way you initially provided it.

Reviewers' comments:

Reviewer's Responses to Questions

**Comments to the Author**

1. If the authors have adequately addressed your comments raised in a previous round of review and you feel that this manuscript is now acceptable for publication, you may indicate that here to bypass the “Comments to the Author” section, enter your conflict of interest statement in the “Confidential to Editor” section, and submit your "Accept" recommendation.

Reviewer #1: All comments have been addressed

Reviewer #2: (No Response)

Reviewer #3: All comments have been addressed

Reviewer #4: All comments have been addressed

2. Does this manuscript meet PLOS Global Public Health’s publication criteria? Is the manuscript technically sound, and do the data support the conclusions? The manuscript must describe methodologically and ethically rigorous research with conclusions that are appropriately drawn based on the data presented.

Reviewer #1: (No Response)

Reviewer #2: (No Response)

Reviewer #3: Yes

Reviewer #4: Yes

3. Has the statistical analysis been performed appropriately and rigorously?

Reviewer #1: (No Response)

Reviewer #2: (No Response)

Reviewer #3: Yes

Reviewer #4: Yes

4. Have the authors made all data underlying the findings in their manuscript fully available (please refer to the Data Availability Statement at the start of the manuscript PDF file)?

Reviewer #1: (No Response)

Reviewer #2: (No Response)

Reviewer #3: Yes

Reviewer #4: Yes

5. Is the manuscript presented in an intelligible fashion and written in standard English?

Reviewer #1: (No Response)

Reviewer #2: (No Response)

Reviewer #3: Yes

Reviewer #4: Yes

6. Review Comments to the Author

Reviewer #1: (No Response)

Reviewer #2: (No Response)

Reviewer #3: The authors have comprehensively addressed the gaps flagged in the initial review

Reviewer #4: Dear authors

Thank you for addressing previous comments. I am attaching a word document with a few track changes, of some items that were missed.

All the best!

7. PLOS authors have the option to publish the peer review history of their article (what does this mean?). If published, this will include your full peer review and any attached files.

**Do you want your identity to be public for this peer review?** For information about this choice, including consent withdrawal, please see our Privacy Policy.

Reviewer #1: **Yes: **Felix Gumaayiri Aabebe

Reviewer #2: No

Reviewer #3: **Yes: **Bonaventure Ahaisibwe

Reviewer #4: No

---

## [Editor Report · Decision Letter 2]

14 Nov 2024

Unimproved Source of Drinking Water and the Associated Factors: Insights from Somalia Health and Demographic Survey 2020.

PGPH-D-24-01613R2

Dear Dr Salih,

We are pleased to inform you that your manuscript 'Unimproved Source of Drinking Water and the Associated Factors: Insights from Somalia Health and Demographic Survey 2020.' has been provisionally accepted for publication in PLOS Global Public Health.

Best regards,

Professor Razak Gyasi, PhD, PD

Academic Editor